# Combining MAS-GiG Model and Related Problems to Optimization in Emergency Evacuation

**Dinh Thi Hong Huyen [1,\*], Hoang Thi Thanh Ha [2] and Michel Occello [3]**

[1] Faculty of Information Technology, Quynhon University, Quynhon 590000, Vietnam
[2] Faculty of Statistics-Informatics, Da Nang University of Economics—The University of Da Nang, Da Nang 550000, Vietnam; ha.htt@due.edu.vn
[3] LCIS, Grenoble Alpes University, 38400 Grenoble, France; michel.occello@lcis.grenoble-inp.fr
[\*] Correspondence: dinhthihonghuyen@qnu.edu.vn

**Abstract:** Emergency evacuation is of paramount importance in protecting human lives and property while enhancing the effectiveness and preparedness of organizations and management agencies in responding to emergencies. In this paper, we propose a method for evacuating passengers to safe places with the shortest possible evacuation time. The proposed method is based on a multi-level multi-agent MAS-GiG model combined with three problems. First, constructing a path map to select the shortest path; second, dividing the space of the experimental environment into smaller areas for efficient management, monitoring, and guiding evacuation; the third, adjusting the speed to handle collision issues and maintain distance between two or more groups of evacuees while moving. We extend our previous study by establishing groups based on the location of passengers and using a MAS-GiG model to guide evacuation. We compare the proposed method with our previous method to provide specific evaluations for the research and research in the future. We tested two methods in the departure hall, first floor, Danang International Airport, Vietnam.

**Keywords:** multi-agent model; emergency evacuation; evacuation guide; group; optimization

## 1. Introduction

Emergency evacuation is a critical issue in emergency management and civilian protection. In emergencies such as fires, an emergency evacuation can save many lives. However, ensuring efficient evacuation remains a complex challenge due to various factors such as limited resources, time constraints, and the unpredictability of emergent situations. Emergency evacuation scenarios often involve large groups of individuals who need to be guided safely and efficiently toward designated safe zones. Traditional evacuation strategies often rely on centralized decision making and predefined evacuation routes, which may not adapt well to dynamically changing conditions. Furthermore, human behavior, group dynamics, and individual relations can significantly impact the success of evacuation efforts. To address these challenges, we proposed a multi-level multi-agent MAS-GiG [1] model combining four related problems as a promising approach. This approach leverages the principles of distributed decision making, self-organization, and coordination among autonomous agents to interact and make decisions to achieve collective goals.

To effectively manage crowds and monitor evacuations, we divide the crowd into smaller subgroups based on the environment structure and the distribution of residents within that environment. This division is carried out to facilitate efficient crowd management and evacuation monitoring. We employ the MAS-GiG model to manage the crowd and monitor evacuations at different levels. Each level encompasses a specific range, with designated representatives responsible for managing the crowd and supervising evacuations within their respective level's scope. We combined the four related problems to answer the following four questions: How can the system manage the crowd and monitor

evacuation at multiple levels? How does the system determine safe routes for passengers to reach exits? How do we address the issue of collisions among evacuating passengers? How do we handle a blocked route or exit during the evacuation process?

To answer the first question, we proposed implementing the MAS-GiG model to manage the system hierarchically and monitor the evacuation process. To determine the scope of the levels within the system, we proposed the Space Partition problem. To address the second question, we have proposed the Constructing Path Graph problem. To address the collision issue in question number 3, we have proposed the Velocity Adjustment problem for the groups. To solve the problem of a blocked route/exit during the evacuation, we have proposed the Evacuation Plan Changing problem. To evaluate the proposed method, we conducted experiments comparing it with the method in [2] to compare the results in order to select the best research method.

Some studies of crowd evacuation include predicting pedestrian movement and emergent behavior [3], finding the route for each pedestrian, avoiding static or dynamic obstacles, and avoiding other pedestrians [4–6]. In [7], the authors planned at global and local levels to enhance route selection and avoid barriers. In [8], the authors proposed a simulation model of pedestrian behavior in different groups and crowd movement, creating individual trajectories for each person in the group. Other studies focus on evacuation planning [9], evaluating and optimizing evacuation plans [10–13], and studying human behavior in emergencies [14,15]. According to [16], the study constructs a BIM—FDS building fire simulation model to reveal the fire smoke dispersion law under the coupling of the typical building structure and fire protection systems. According to [17], the authors proposed a new way for a smart building fire evacuation and control system based on the IoT to direct individuals along an evacuation route during fire incidents efficiently. In [18], the co-effectiveness of stairwells, walkways, and room doors in reducing total evacuation time was investigated using simulation and machine learning. The authors selected a typical high-rise teaching building as an example and integrated two simulation software, Pyrosim and Pathfinder, to compare the available safe evacuation time (ASET) and required safe evacuation time (RSET). In the paper [19], the authors presented a case study on the behavior of fire in a sealed pressurized hotel building located in Tibet. A fire dynamics simulator (FDS) is used to study fire behavior under different design fire scenarios. The available evacuation time under different fire scenarios was obtained by analyzing the oxygen concentration, gas content, temperature, visibility, and other indicators. This study supports a fire protection design scheme. According to [20], the authors focused on the analysis of fire smoke spread, visibility, smoke temperature distribution, and variation curves in an atrium.

The impact of fires on the evacuation process [17–20], and the characteristics and behavior of humans during evacuation [21], has been tested. In [22], the authors focused on evacuation plans for buildings with multiple escape routes based on a network model of indoor pathways. According to [23], the authors proposed an agent-based model that provides a framework for exploring individual, social, and technological aspects in crowd evacuation.

The remainder of the article is structured as follows: Section 2 describes the problem; Section 3 presents the proposed method; Section 4 details the experiment and an evaluation of the results; Section 5 is the discussion; and, finally, there is a conclusion.

## 2. Description of the Problem

To optimize emergency evacuation, we studied the problem of guiding an emergency evacuation when there is a fire in the lounge, departure hall, and first floor of Danang International Airport. After checking in and passing through the security gate, passengers enter the lounge, where a large number of passengers are usually concentrated at a relatively high density. When a fire appears at a location in a lounge, how do we guide the evacuation of all passengers to a safe place as quickly as possible? We aim to find a way to evacuate N people from the starting points to the destination points (safe places) so that the total travel

time is minimized. The problem is described as an optimization problem with notations, constraints, and objective functions as follows.

The notations are used to describe the optimization problem (see Table 1).

**Table 1.** The notations used in problem description.

| Notations | |
| --- | --- |
| $N$ | The total number of people who need to be evacuated |
| $n$ | The total starting locations |
| $m$ | The total number of destinations (safe places) |
| $i$ | The $i$-th starting location |
| $j$ | The $j$-th destination |
| $N_i$ | The number of people to be evacuated from starting location $i$ |
| $N_j$ | The number of people to be evacuated to destination location $j$ |
| $c$ | The capacity of each road segment is limited to a certain number of people |
| $d_s$ | The length of road segment $s$ |
| $v_s$ | The velocity of an evacuee on a road segment $s$ |
| $t_s$ | The time it takes for an evacuee to travel the entire distance $d_s$ at the speed $v_s$ |

● Constraints

- Each person must be evacuated from a starting location to a destination location.

$$\sum_{j=1...m} x_{ij} = 1, \text{ với } i = 1, 2, \ldots, n \tag{1}$$

- The total number of evacuees at a given time must not exceed the capacity of the evacuation route.

$$\sum_{j=1...m} x_{ij} \leq c, \text{ với } i = 1, 2, \ldots, n \tag{2}$$

- The number of evacuees from each starting location must not exceed the number of evacuees present at that location.

$$\sum_{j=1...m} x_{ij} \leq N_i, \text{ với } i = 1, 2, \ldots, n \tag{3}$$

- The number of evacuees to each destination location must not exceed the number of evacuees who need to be evacuated to that location.

$$\sum_{i=1...n} x_{ij} \leq N_j, \text{ với } j = 1, 2, \ldots, m \tag{4}$$

● Decision variables

The decision variables in emergency evacuation problems are $x_{ij}$, with $i = 1, 2, \ldots, n$ and $j = 1, 2, \ldots, m$. The variable $x_{ij}$ is a binary variable that determines whether the passenger at starting location ith is evacuated to the safe location $j$-th or not. If yes, then $x_{ij} = 1$, otherwise $x_{ij} = 0$.

Objective function: Minimize the total evacuation time.

$$\min \sum_{i=1...n} \sum_{j=1...m} t_{ij} \times x_{ij} \tag{5}$$

with $t_{ij}$ being the evacuation time for an evacuee to move from a location $i$ at the lounge to a safe location $j$.

## 3. Proposed Method

### 3.1. The Problem-Solving Approach Based on the Modeling

When a fire occurs in a lounge, passengers will choose the nearest exit to evacuate. However, the majority of passengers are not aware of the exits in the lounge, a large number

of passengers are in a narrow space, the pathways have limitations, and congestion can easily occur at corners or intersections of paths or safe exits, reducing the evacuation speed, prolonging the total evacuation time, and increasing the risk to people.

The issues to be addressed in the optimization problem include identifying the best evacuation route; dividing the area appropriately for managing, monitoring, and guiding evacuation at different levels based on the multi-level multi-agent model MAS-GiG; and adjusting the movement speed of groups when there is a collision or a gap between adjacent groups.

To solve the optimization problem, we propose a method that combines modeling and simulating crowds based on the multi-level multi-agent MAS-GiS model and supporting problems such as space division, constructing a path graph, and adjusting the moving speed of groups. The MAS-GiS model coordinates and guides the evacuation of all passengers to safe locations on the shortest routes, ensuring that the total evacuation time is minimized. This model is experimented in a 2D space, where each position in space has coordinates $(x_i, y_i)$, the length of the path $i$ is $d_i$, the moving speed of each agent representing an evacuee is $v_i$, and the time it takes for an agent to move along path $i$ with speed $v_i$ is $t_i$. The method ensures Equations (1)–(4) and optimizes the objective Equation (5), which means evacuating all passengers to safe locations in a way that minimizes the total evacuation time.

### 3.2. The Multi-Level Multi-Agent MAS-GiS Model

We propose the multi-level multi-agent MAS-GiS model to guide passenger evacuation by level. According to reference [1], the MAS-GiS model is a combination of the AEIO architecture [24] and the multi-level group model [25].

#### 3.2.1. AEIO Architecture

An AEIO-based multi-agent model consists of four components: A—the agents, E—the environment, I—the interactions, and O—the organization.

- A: Agents

Each agent represents an individual residing in the environment; it exhibits autonomous, reactive, and interactive behavior. It is characterized by a set of properties P, knowledge K, and a set of roles R. It acts in the environment through a set of actions A.

- E: Environment

The environment is the shared space where the agents are present. The environment consists of objects distributed within it and the relationships between them. These objects either affect the agents or they are affected by them. In the application, environmental objects include seats, walkways, shops, restrooms, entrances, exits, etc.

- I: Interactions

Interactions in the application include interactions on the same level (horizontal) and interactions on different levels (vertical). Horizontal interaction is the interaction between the group representative and the group members and interactions between group members themselves. A group representative interacts with another group representative of the same level. Vertical interaction is the interaction between a group representative and a group representative at a higher or lower level. Interactions are performed by sending/receiving messages.

The structure of a message in the MAS-GiG model for the application is based on the interaction protocol in MASH [26], which is a two-layer message and frame interaction mechanism. This mechanism is similar to the two layers of Network and Datalink in the seven-layer model of the OSI computer network [27].

- O: Organization

The organization in the MAS-GiG model for the application includes group structure and relationships. In a group, there is a representative and members. A group member at a level is a representative of a group at a lower level adjacent to it.

### 3.2.2. Multi-Level Group Model

The multi-level group model is formed by a bottom-up mechanism and is described as follows:

- Level 0: Agents at level 0 are called basic agents, representing a passenger in the application. At this level, there is no organization or group structure.
- Level 1: This is the first group level, where each group member is an agent at level 0. Agents belong to the same group when they are within the same range.
- Level $n$ ($n \geq 2$): This is the $n$-th group level, where each group member is a representative of a group at level ($n - 1$). Agents at level $n$ belong to the same group when they are in the same range.

### 3.2.3. The Formation of Levels

The formation of levels in the MAS-GiG model is carried out as follows:

- Level 0: Agents at level 0 are called basic agents, representing a passenger in the application.
- Level 1: A group at this level is formed from agents at level 0, such that they are in the same range r.
- Level $n$ ($n \geq 2$): Each agent at level n represents a group at level ($n - 1$). The group at level $n$ is formed according to Equation (6).

$$\begin{cases} G_{(n-1)i(i=1...g)} \overset{Representatives}{\longleftarrow} I_{ni\ (i=1...m)} \\ G_{nk(k=1...g)} = \left\{ \left( I_{ni(i=1...m)},\ I_{nj(j=1...m)} \right) : I_{ni} r\ I_{nj} \right\} \end{cases} \tag{6}$$

where $r$ is the range in which agents belong to the same group. For each $I_{ni}$, $I_{nj}$ is the representative of the groups $G_{(n-1)i}$, $G_{(n-1)j}$ at level ($n - 1$), $m$ is a constant representing the number of agents at level ($n - 1$), and g is the number of groups at level $n$.

### 3.2.4. Choose Group Representative

The selection of a group representative is based on the perception score of each agent. The perception score includes knowledge about the environment—$a_{ek}$, personal experience in emergency evacuation—$a_{pe}$, and decision-making time—$a_{dt}$. The agent with the highest perception score in the group is chosen as the group representative. The perception score of each agent is calculated according to Formula (7).

$$K_a = a_{ek} + a_{pe} + a_{dt} \tag{7}$$

### 3.3. Four Supporting Problems

We proposed four support problems: The first problem is the Space Partition problem. The second problem is the Constructing Path Graph problem. The third problem is the Velocity Adjustment problem for groups. The fourth problem is the Changing Evacuation Plans problem.

### 3.3.1. The Space Partition Problem

The Space Partition problem of environment E is to divide it into multiple subspaces to determine the levels of the MAS-GiG model. In the application, the range of the departure hall includes lounges, each of which is divided into zones, and each zone is divided into smaller areas. The purpose of this partition is to apply the multi-level multi-agent MAS-GiG model to the application for managing, monitoring, and guiding passenger evacuation at different levels, to achieve high efficiency in evacuation and increase flexibility when there are unexpected changes in the evacuation plan.

The space is denoted as *SE*. The subspaces are independent spaces, and they have a topological relationship with each other, denoted as $S_{ai}$ ($i = 1, 2, 3, \ldots, n$). The subspaces $S_{ai}$ satisfy the following conditions:

$$SE = S_{a1} \cup S_{a2} \cup \cdots \cup S_{ai} \cdots \cup S_{an}, \ n \geq 0 \tag{8}$$

$$S_{ai}, \ S_{aj} \subset SE : \ S_{ai} \bigcap S_{aj} = \varnothing, \ i \neq j \ \text{v} \ i,j \in \{1,2,3,\cdots,n\} \tag{9}$$

This means that the $S_{ai}$ subspaces are subsets of the *SE* space (8), and these subspaces do not intersect each other (9).

In the application with the scope of a lounge, the space is the lounge itself, and it is divided into smaller areas for evacuation planning. For example, *Lounge*5 can be divided into three areas, *Area*51, *Area*52, and *Area*53, such that:

$$\begin{cases} Lounge5 = Area51 \cup Area52 \cup Area53 \\ Area51, \ Area52, \ Area53 \subset Lounge5 \\ Area51 \cap Area52 \cap Area53 = \varnothing \end{cases}$$

### 3.3.2. The Constructing Path Graph Problem

To determine the shortest path from a position of a group of passengers to an exit, we rely on the structure of the departure hall to construct a path graph G = <V, E>, where V is the set of vertices, and E is the set of edges of the graph. From the path graph, we can determine the shortest path from one vertex to another vertex of G using Dijkstra's algorithm. Based on the spatial structure of the departure hall, including corridors, routes, and seats, a graph of paths is created. There are two corresponding data structures, including the road and road segment, to store these network data. These data can be considered raw data that need to be refined through an algorithm to find the vertices of the path graph, which are the intersection points between roads or the dead ends of a road, and the adjacent points to create the edges of the graph. This graph is a directed graph.

### 3.3.3. The Velocity Adjustment Problem for Groups

If two adjacent groups collide, the procedure of reducing the speed of groups from the collided group to the last group in the area is performed. The process of reducing speed is carried out until there are no more collisions between groups. If there is a gap between group i and group $i + 1$ when moving towards the exit, the procedure of increasing the speed of groups from group $i + 1$ to the last group in the area is performed, subject to the condition that $v_j <= v_l$ ($j = i + 1 \ldots m$), where m is the last group in the area, and $v_l$ is the limited speed. If $v_j > v_l$ ($j = i + 1 \ldots m$) and there are still groups in the area that have not been adjusted for speed, the procedure of reducing speed is performed. The process is carried out until all groups in the area have been adjusted for speed (Figure 1).

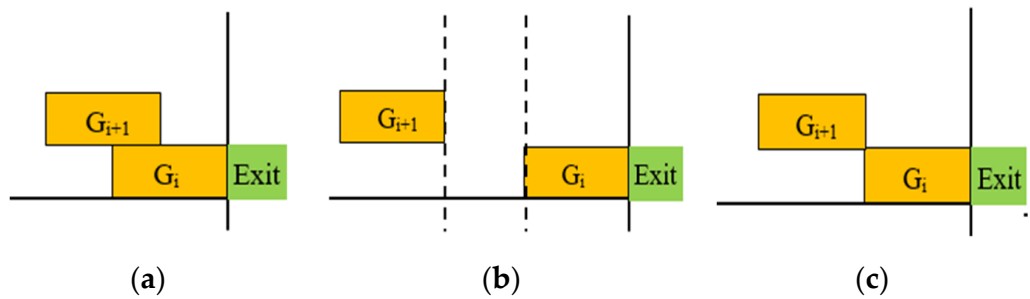

**Figure 1.** Illustrates three cases that can occur between two groups when moving. (**a**) Shows two groups colliding with each other. (**b**) Shows two groups with a distance between them. (**c**) Shows two groups that are not in a collision, and there is no distance between them.

3.3.4. The Changing Evacuation Plans Problem

When passenger groups are executing the evacuation plan, and if a route is blocked, cease the evacuation plan of the passenger groups that move along this route; the airport operator establishes a new evacuation plan based on the current positions of the groups and the exit. Then, the operator sends the new evacuation plan to the lounge's representatives. When the lounge's representatives receive the new plan, they send the new evacuation plan to the area's representatives. When the area's representatives receive the new evacuation plan, they send it to the group's representatives, and when the group's representatives receive the new evacuation plan, they instruct the members of their groups to evacuate according to the new evacuation plan.

When passenger groups are executing the evacuation plan, and an exit is blocked, stop the evacuation of passenger groups moving towards that exit; the airport operator establishes a new evacuation plan based on the current positions of the groups and nearby safe exits. Subsequently, the new evacuation plan is deployed to the lower levels similarly as in the case of a blocked route.

*3.4. Applying the MAS-GiG Model to the Problem of Evacuating Passengers in a Lounge*

3.4.1. Modeling the Problem of Evacuating Passengers Using the AEIO Architecture

The AEIO architecture is applied to the evacuation problem as follows:

- A consists of a set of agents: an agent representing a person residing in the departure hall such as passengers, airport staff, and emergency management system staff of the airport.
- E consists of a set of objects in the lounge, such as chairs, walkways, shops, restaurants, restrooms, entrances to the waiting area, aircraft exits, etc.
- I include interactions between agents with each other and interactions between agents and the environment. The interactions between agents include asking about the flight, going shopping together, asking about the restaurant, etc. Interactions between agents and the environment include agents choosing seats and waiting for the flight, dining at a restaurant, shopping at a store, listening to announcements or watching information on TV, viewing information on bulletin boards, etc.
- O includes group structure and relationships. Each group in the MAS-GiG model has a group representative and group members. The relationship within a group includes the relationship between the group representative and the group members and the relationship between the group members themselves, which is the same group relationship. For example, passengers who are close to each other are grouped; they share a group, have the same evacuation plan, and follow the instructions of the group representative. The same-level relationship is the relationship between different groups at the same level. For example, passenger groups in the same area have the same area level relationship with each other. The group structure at the area level is similar to the group level; each area also has a representative and members. Each individual in an area is a representative of a group at the group level. Individuals in an area share the same area representative, they evacuate through the same exit route, etc.

3.4.2. Determine the Number of Levels in the MAS-GiG Model for the Evacuation Problem

Based on the spatial structure of the departure hall and the data on passenger distribution in the lounge, we propose the number of levels of the MAS-GiG model for the application. There are five levels: the individual level (Indie), group level (Group), area level (Area), lounge level (Lounge), and airport operator level (AirportOperator).

- Level 0 is individual level (Indie). At this level, each agent represents a passenger; they have an equal role.
- Level 1 is the first group level; this is the Group level. Agents in the same group are within the same range r. Each group has a group representative and group members. The group representative is the GroupLeader.

- Level 2 is the second group level, which is the Area level. Each agent at this level is a representative of a group at the group level. The area representative is Guide.
- Level 3 is the third group level, which is the Lounge level. Each agent at this level is a representative of a group at the Area level. The lounge representative is GuideLeader.
- Level 4 is the fourth group level, which is the operational level. Each agent at this level represents a group at the lounge level. The representative for the emergency management system operating department is called the Airport Operator (AO).

3.4.3. Perform the Passenger Evacuation Problem Using the Proposed Method

First, passengers are grouped based on their position distance, with one representative selected for each group using Formula (7). Then, based on the passenger distribution ratio, fire data, and lounge structure, the areas for the groups are determined according to the first problem using Formulas (8) and (9). Similar to the groups, each area has a representative selected using Formula (7) from the group representatives. Each lounge also has a representative determined using Formula (7) from the area representatives. Groups within the same area move towards the same exit. The evacuation instruction is carried out according to the plan set by the management level and implemented to the lower levels. Based on their role, each level representative performs corresponding actions.

Second, all evacuation groups in the same area are moved in order of the shortest path from their position to the exit. The shortest path is determined based on the second problem. In the case that passengers from two or more areas need to exit from the same exit, the movement order is similar to the order of movement of the groups, meaning that the area closest to the exit moves first. In the case of two groups colliding, the velocity reduction algorithm from the third problem is used to solve the collision problem. In the case there is a gap between two groups, the velocity increase algorithm from the third problem is used.

Figure 2 illustrates lounge 5 represented by Guideleader5. Room 5 has three areas, A51, A52, and A53. Area A51 has exit E1 and is represented by Guide51, with passenger groups G11 to G18. Area A52 has exit E2 and is represented by Guide52, with passenger groups G21 to G26. Area A53 has exit E3 and is represented by Guide53, with passenger groups G31 to G39. The evacuation plan is executed in the three areas, with the order of movement for each group based on the distance to their area's exit. In area A51, groups move in the order of G11, G12, G14, G13, G15, G16, G17, and G18 with corresponding distances to exit E1 of 2, 3, 6, 8, 13, 14, 15, and 16, respectively. In area A52, groups move in the order of G21, G22, G23, G24, G25, and G26 with corresponding distances to exit E2 of 3, 4, 6, 7, 9, and 10, respectively. In area A53, groups move in the order of G31, G32, G33, G34, G35, G36, G37, G38, and G39 with corresponding distances to exit E3 of 1, 2, 4, 5, 7, 8, 10, 12, and 13, respectively. Each group follows the shortest path guided by their respective group leaders.



**Figure 2.** Abstract illustration of lounge 5.

## 4. Experimentation and Result Evaluation

### 4.1. The Passenger Data

Passenger distribution data in the lounge are based on the distribution ratio in the document [28]. The distribution of passengers enters the lounge three times, 6:00 a.m., 10:00 a.m., and 6:00 p.m., within the time of 2 h and 50 min to 1 h before takeoff. We assume that approximately 300 passengers are distributed to each lounge based on the passenger distribution ratio at 6:00 a.m., and the location of each passenger is randomly assigned.

### 4.2. Environmental Parameters

According to the document on emergency evacuation standards from the National Institute of Standards and Technology of the United States [29], the size of the emergency exit doors is 45 pixels, representing the real size of 3 m. We assume that 100% of the evacuees are normal, able-bodied individuals with awareness and good health.

### 4.3. Experiment

We experimented with the proposed method based on the simulation framework MASH [25]. Based on this framework, we developed the application, implemented the program, and conducted experiments on the problem of passenger evacuation at the departure hall, first floor, Danang International Airport, Vietnam (Figure 3).

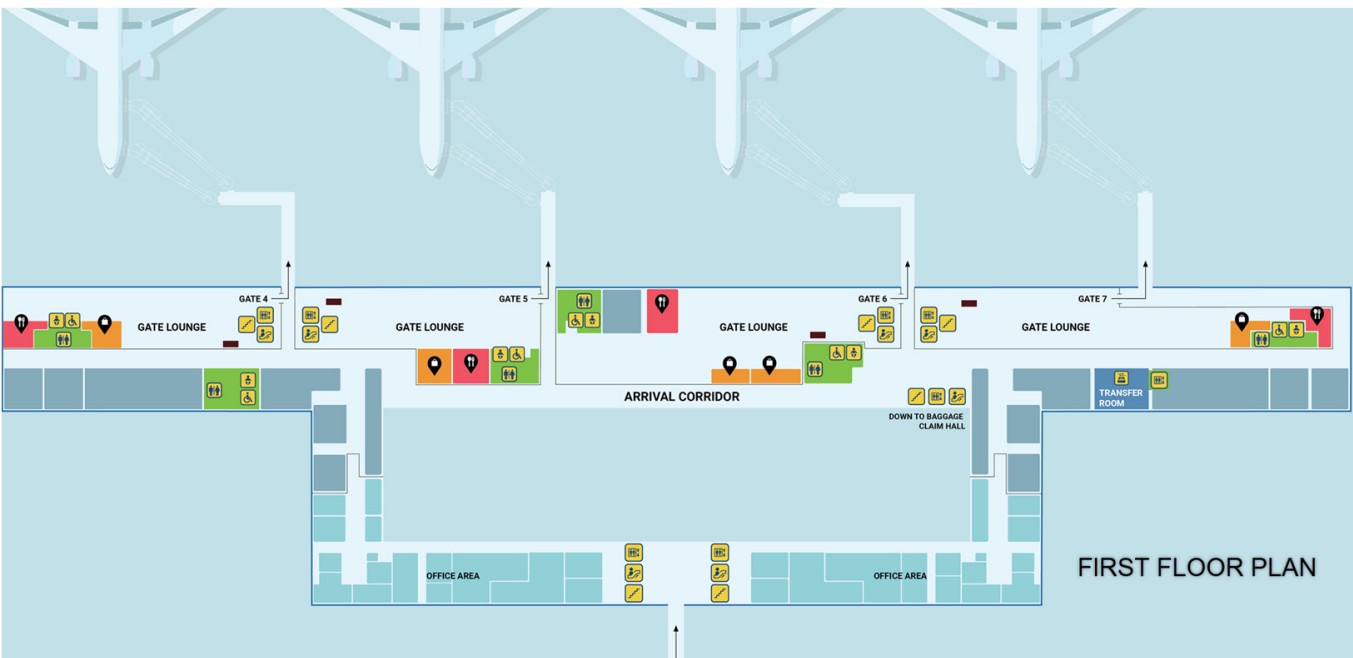

**Figure 3.** Departure hall map.

The departure hall has four lounges, from lounge 4 to lounge 7. Each lounge has a similar structure, including an entrance to the lounge, a door to the airplane, and an exit to the common corridor. The division of areas within each lounge is based on their actual structure, such as sitting areas, shops, restaurants, restrooms, pathways, and entrances. There are three doors used as exits which are the door to the common corridor, the entrance to the lounge, and the door to the airplane.

Each lounge has its own evacuation plan. The total evacuation time of each lounge is calculated by the total evacuation time of the exit with the longest total evacuation time. The total evacuation time of the departure hall is equal to the total evacuation time of the lounge with the longest total evacuation time.

### 4.3.1. Testing within the Scope of One Lounge

A brief description of the symbols in Figure 4 includes the safe area (1), exit door to board the aircraft (2), lounge entrance (3), exit to the common corridor (4), waiting area (5), shop (6), restaurant (7), and restroom (8). The exit to the common corridor (4), exit door to board the aircraft (2), and lounge entrance (3) are three exits E1, E2, and E3. The three fire locations are F1, F2, and F3.

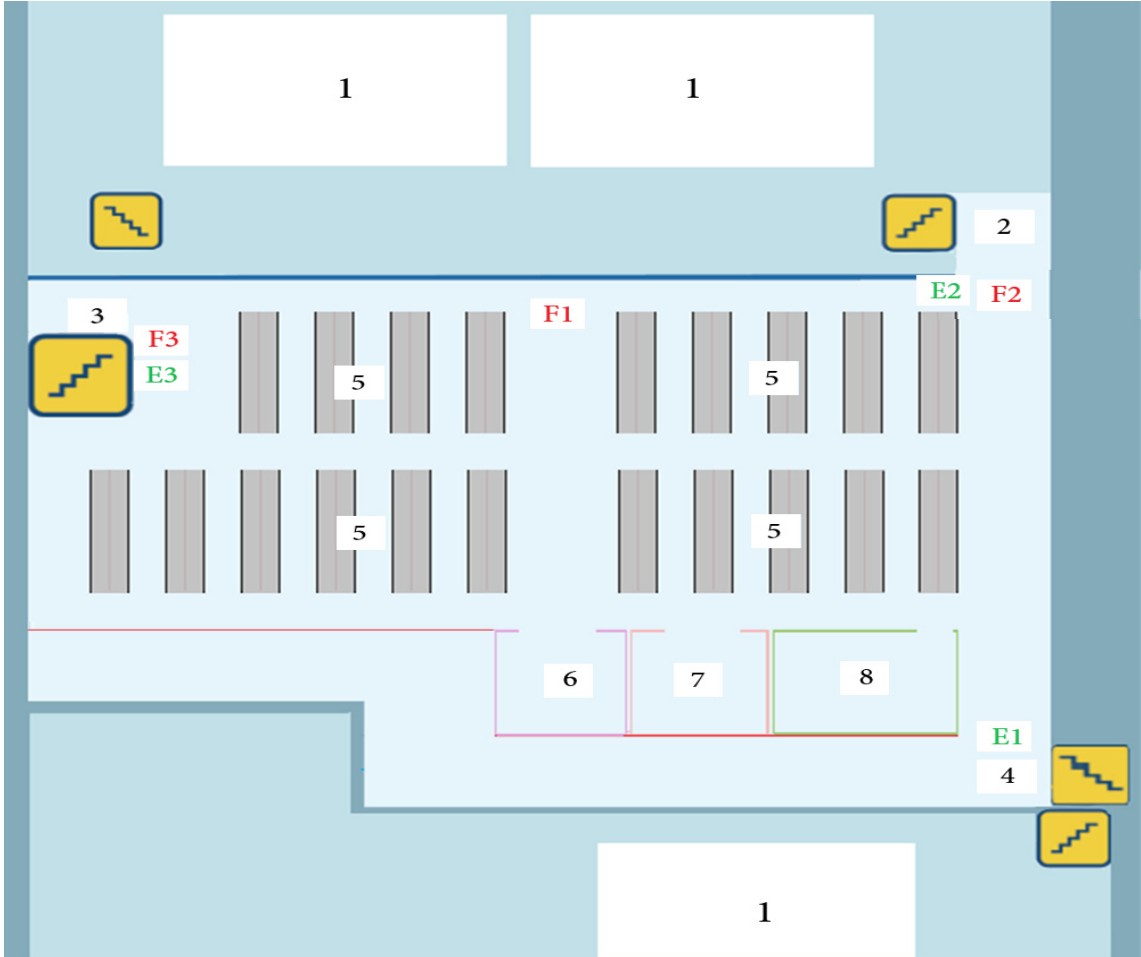

**Figure 4.** The structure of lounge 5.

We experimented with lounge 5 using the proposed method, where the speed for each agent was around 1.12 m/s. The total number of agents in the lounge was about 300. The experiment results are illustrated with three scenarios: scenario 1, fire at a location not coinciding with any exit (F1); scenario 2, fire at the board the aircraft (F2); and scenario 3, fire at the entrance to the lounge (F3).

- Scenario 1

As the fire occurred at F1, all three exits were used to evacuate passengers. The simulation interface for scenario 1 is displayed in Figure 5.

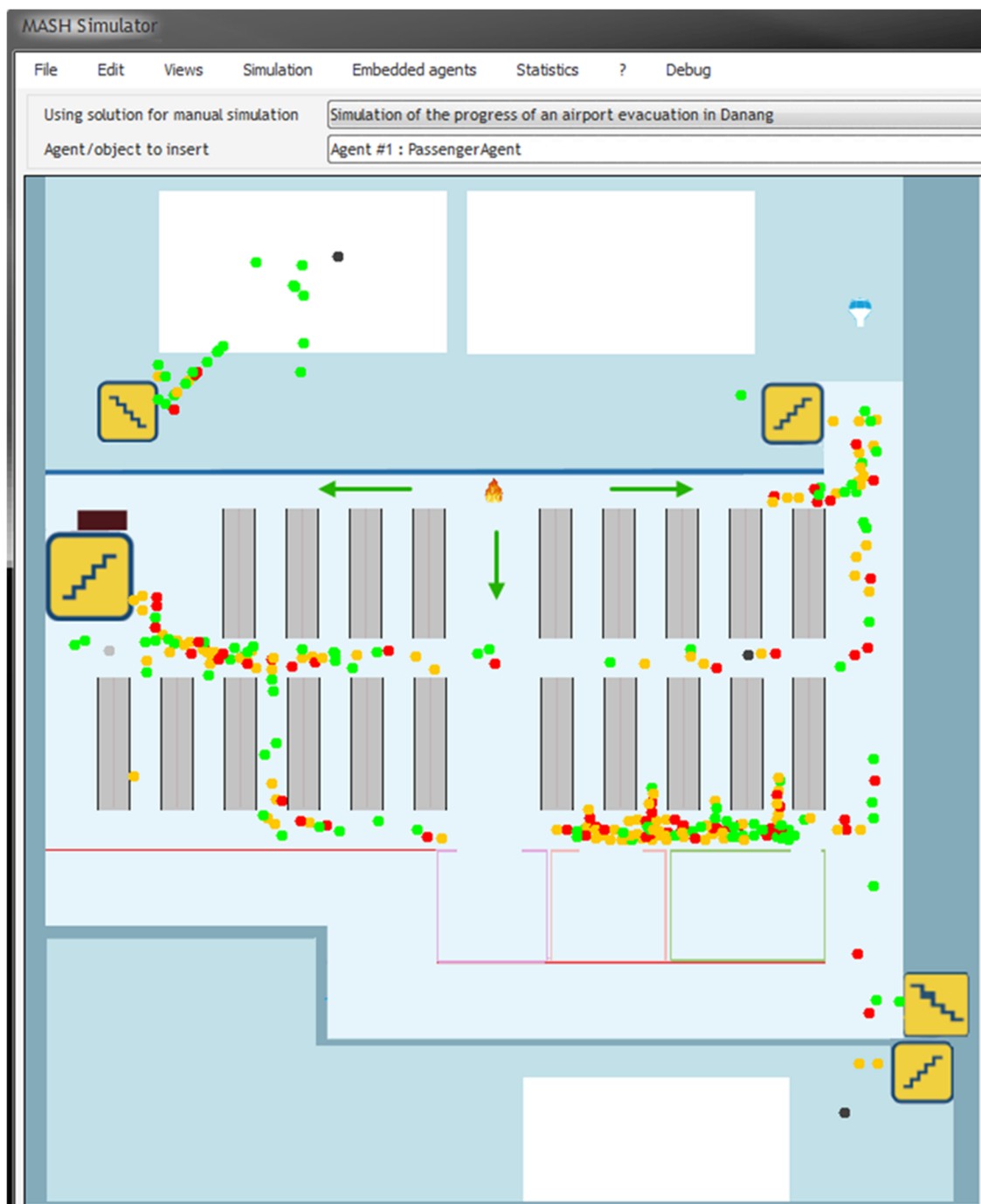

**Figure 5.** The simulation interface for scenario 1.

The experimental results for scenario 1 are summarized in Table 2. The total evacuation time for lounge 5 in scenario 1 was 133 s.

**Table 2.** The experimental results for scenario 1.

| Parameters | Scenarios<br>The Experimental Results in<br>Lounge 5 | E1 | E2 | E3 |
|---|---|---|---|---|
| Total evacuation time | 133 | 95 | 107 | 133 |
| Total number of agents arrived safe places | 297 | 87 | 86 | 124 |

- Scenario 2

The fire occurred at F2, which is the aircraft exit door in the lounge. At that time, passengers can only be evacuated to the remaining two exits, E1 and E3. The simulation interface for scenario 2 is displayed in Figure 6.

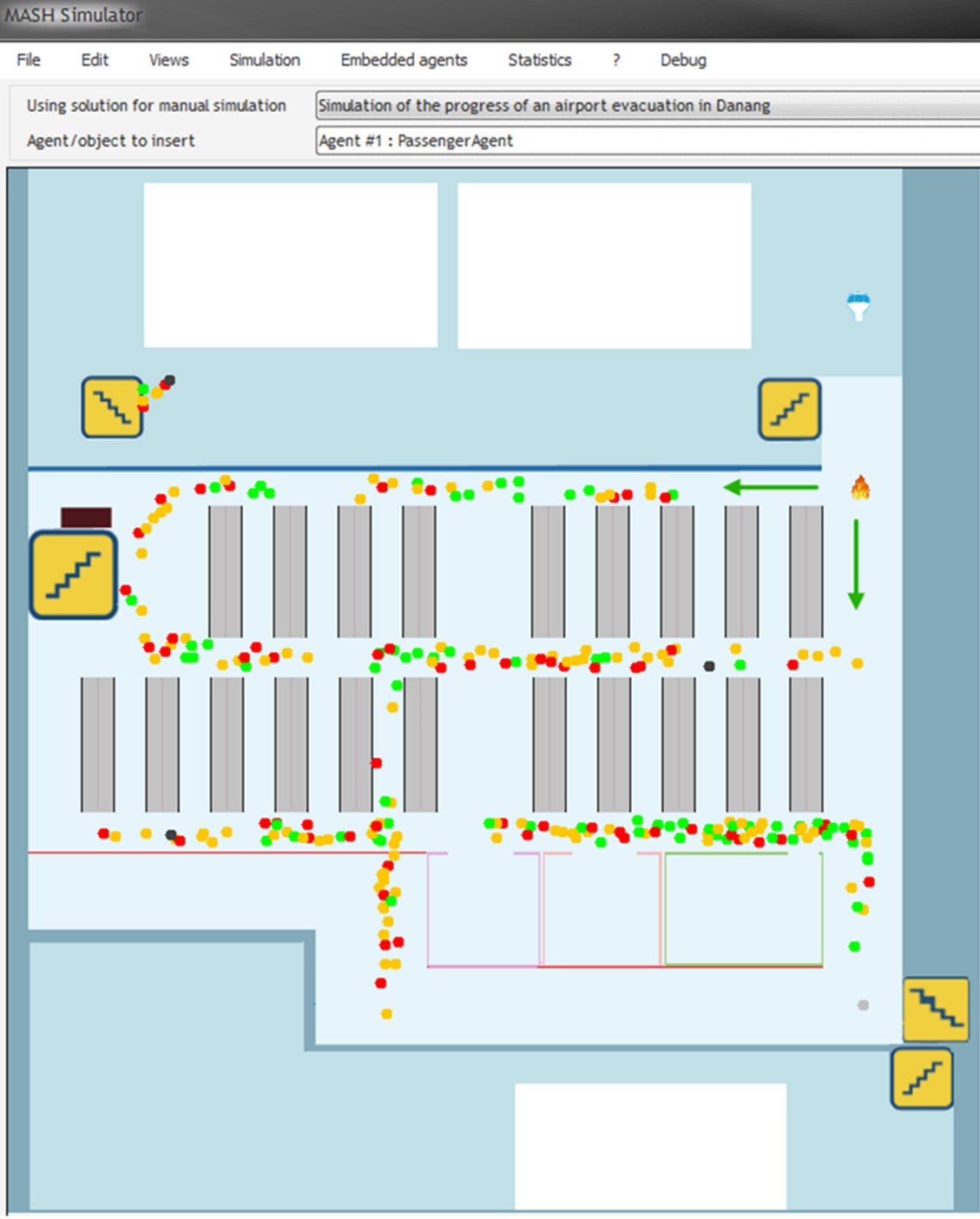

**Figure 6.** The simulation interface for scenario 2.

The experimental results for scenario 2 are summarized in Table 3. The total evacuation time for lounge 5 in scenario 2 was 197 s.

**Table 3.** The experimental results for scenario 2.

| Parameters \ Scenarios | The Experimental Results in Lounge 5 | E1 | E2 | E3 |
|---|---|---|---|---|
| Total evacuation time | 197 | 197 | 0 | 133 |
| Total number of agents arrived safe places | 291 | 175 | 0 | 116 |

- Scenario 3

The fire occurred at F3, which is at the entrance to the lounge. At that time, passengers can only be evacuated to the remaining two exits, E1 and E2. The simulation interface for scenario 3 is displayed in Figure 7.

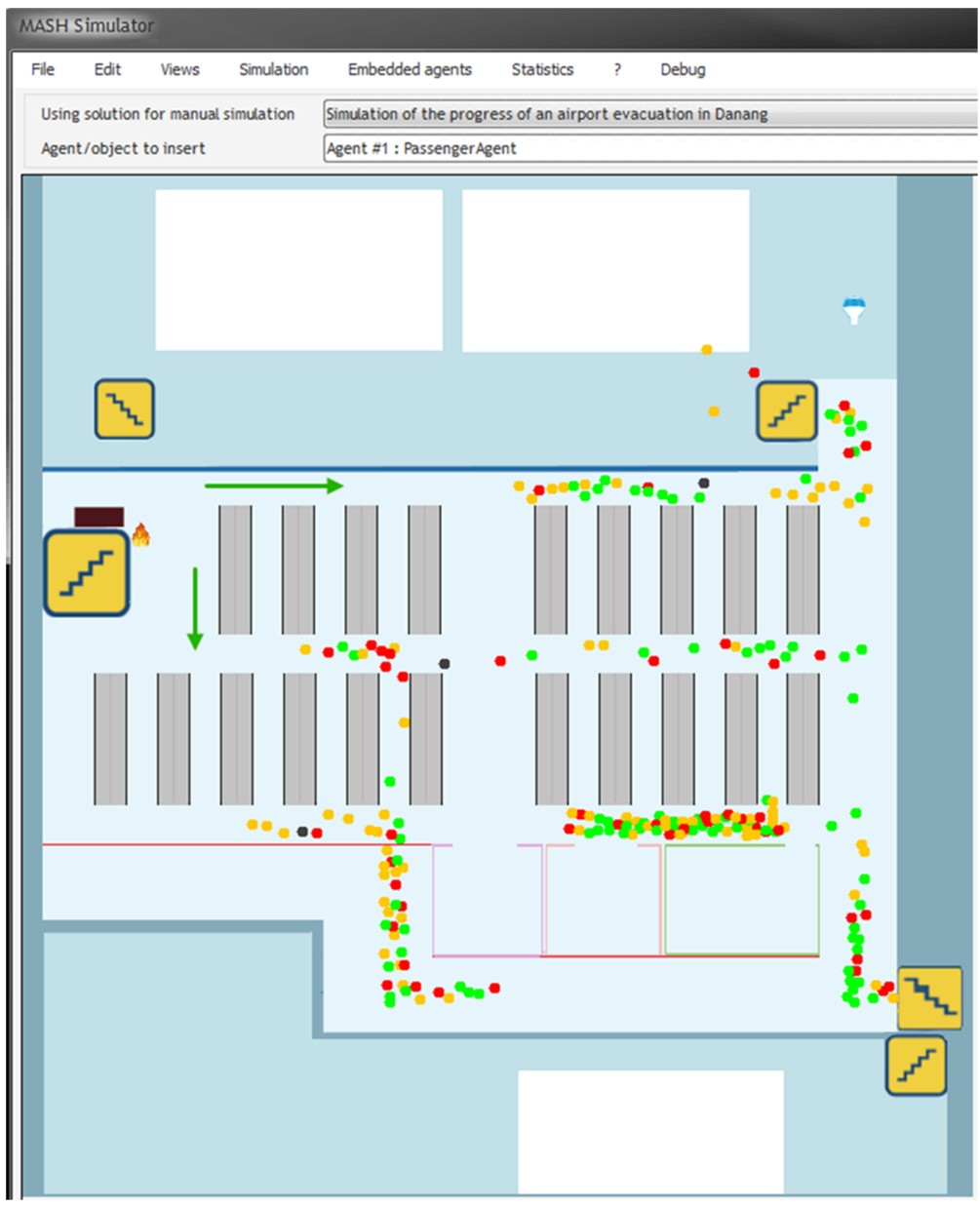

**Figure 7.** The simulation interface for scenario 3.

The experimental results for scenario 3 are summarized in Table 4. The total evacuation time for lounge 5 in scenario 3 was 189 s.

**Table 4.** The experimental results for scenario 3.

| Parameters      Scenarios | The Experimental Results in Lounge 5 | E1 | E2 | E3 |
|---|---|---|---|---|
| Total evacuation time | 189 | 189 | 159 | 0 |
| Total number of agents arrived safe places | 291 | 161 | 130 | 0 |

4.3.2. Testing within the Scope of Four Lounges

Similar to the experiment within the scope of a lounge, we conducted experiments for four lounges. We tested the proposed method and the method in the document [2] by conducting ten trials for each method. The number of agents generated in each lounge was 300 agents. The average movement speed of agents for both methods was around 1.12 m/s. We calculated the average total evacuation time for each method for each scenario.

Assuming a fire occurs in lounge 5, there are three scenarios corresponding to three fire locations (F1, F2, F3) similar to the experimental case within the scope of a lounge. For scenarios 2 and 3, the evacuation plan for lounge 5 only evacuates passengers through the two exits that are not on fire. For the remaining lounges, the evacuation plan for each lounge uses all three exits for each scenario.

- Scenario 1

The fire occurs at location F1 of lounge 5. In this case, all four lounges use three exits to evacuate passengers. The experimental results of the two methods are shown in Table 5.

**Table 5.** The experimental results of the two methods for scenario 1.

| Methods      Scenarios | The Average Total Evacuation Time of the Departure Hall | L4 | L5 | L6 | L7 |
|---|---|---|---|---|---|
| MAS-GiG [2] | 199 | 157 | 199 | 192 | 190 |
| Proposed method | 197 | 159 | 197 | 190 | 194 |

- Scenario 2

The fire occurs at location F2 of lounge 5. In this case, the evacuation plan for lounge 5 only uses two exits, E1 and E3. As for the remaining three lounges, the evacuation plan for each lounge uses all three exits to evacuate passengers. The experimental results of the two methods for Scenario 2 are shown in Table 6.

**Table 6.** The experimental results of the two methods for scenario 2.

| Methods      Scenarios | The Average Total Evacuation Time of the Departure Hall | L4 | L5 | L6 | L7 |
|---|---|---|---|---|---|
| MAS-GiG [2] | 205 | 203 | 198 | 205 | 201 |
| Proposed method | 206 | 206 | 195 | 201 | 199 |

- Scenario 3

The fire occurs at location F3 of lounge 5. In this case, the evacuation plan for lounge 5 only uses two exits, E1 and E2. As for the remaining three lounges, the evacuation plan for each lounge uses all three exits to evacuate passengers. The experimental results of the two methods for Scenario 3 are shown in Table 7.

**Table 7.** The experimental results of the two methods for scenario 3.

| Methods      Scenarios | The Average Total Evacuation Time of the Departure Hall | L4 | L5 | L6 | L7 |
|---|---|---|---|---|---|
| MAS-GiG [2] | 207 | 207 | 204 | 194 | 203 |
| Proposed method | 205 | 205 | 202 | 199 | 194 |

The experimental results of the three scenarios show that the average of the total evacuation time of the proposed method is equivalent to the method in the paper [2].

*4.4. Evacuation*

4.4.1. Comparison

The proposed method based on the MAS-GiG model to resolve optimization in emergency evacuation and the study on a bi-objective robust possibilistic cooperative gradual maximal covering model for relief supply chain with uncertainty are two different research studies that address optimization problems in different contexts. The following are some comparisons between the two methods based on several criteria:

- Context

MAS-GiG model for emergency evacuation: This study focuses on emergency evacuation scenarios, where the goal is to optimize the evacuation process using a multi-level multi-agent MAS-GiG model. It aims to find the best routes, allocate resources, and coordinate the evacuation process efficiently.

Bi-objective robust possibilistic cooperative gradual maximal covering model for relief supply chain: This study focuses on the relief supply chain in uncertain environments. It aims to optimize the allocation of relief supplies to affected areas considering multiple objectives, such as maximizing the coverage and minimizing the cost. The model incorporates robustness and possibilistic approaches to handle uncertainties in demand and supply.

- Optimization Objectives

MAS-GiG model for emergency evacuation: The primary objective is to optimize the emergency evacuation process, considering factors such as minimizing evacuation time, maximizing safety, and efficiently allocating resources.

Bi-objective robust possibilistic cooperative gradual maximal covering model for relief supply chain: The objectives are to maximize the coverage of relief supplies to affected areas and minimize the cost of the supply chain operation. The model aims to find a balance between these two objectives while considering uncertainties in the environment.

- Decision-Making Approaches

MAS-GiG model for emergency evacuation: The MAS-GiG model utilizes a cooperative approach where multiple agents collaborate and make decentralized decisions to achieve the overall optimization objective. The model adapts to dynamic environments and considers the safety and efficiency of the evacuation process.

Bi-objective robust possibilistic cooperative gradual maximal covering model for relief supply chain: The relief supply chain model utilizes a cooperative and possibilistic approach to handle uncertainties. It considers both robustness and possibilistic measures to ensure the reliability and flexibility of the supply chain in uncertain situations.

- Application Domains

MAS-GiG model for emergency evacuation: This model is specifically designed for emergency evacuation scenarios, such as fire, natural disasters, terrorist attacks, or any situation requiring the rapid and efficient evacuation of people.

Bi-objective robust possibilistic cooperative gradual maximal covering model for relief supply chain: This model focuses on optimizing the relief supply chain operations in uncertain environments, where demand and supply conditions are subject to variations and uncertainties. It can be applied in disaster response and humanitarian logistics contexts.

In summary, the MAS-GiG model for emergency evacuation and the bi-objective robust possibilistic cooperative gradual maximal covering model for relief supply chain with uncertainty are two distinct research studies that address optimization problems in different contexts. While the MAS-GiG model focuses on optimizing the emergency evacuation process using a multi-agent system approach, the relief supply chain model aims to optimize the allocation of relief supplies in uncertain environments. Both studies contribute to enhancing decision making and optimization in their respective domains.

4.4.2. The Usefulness of RFID for Emergency Evacuation

According to [30–33], the usefulness of RFID (radio frequency identification) in emergency evacuation can be presented as follows:

Location tracking and monitoring: RFID enables accurate tracking and monitoring of personnel, equipment, and critical assets during the evacuation process. This facilitates the efficient management and coordination of the evacuation.

Efficient identification: During an emergency evacuation, time is of the essence. RFID technology enables the quick and accurate identification of individuals and resources.

Coordination and guidance: RFID can be used to send messages and instructions to each participant during the evacuation. Through RFID devices, individuals can receive specific information about evacuation routes, safe locations, and necessary instructions for safe and efficient movement.

Control and monitoring: RFID systems allow comprehensive control and monitoring of the evacuation process. By tracking the location and status of each participant, the system can detect and address issues that may arise during the evacuation, such as congestion, collisions, or emergencies.

Resource optimization: RFID assists in optimizing resource utilization during emergency evacuation. By tracking and managing information about critical assets and resources, the system can provide the optimal allocation and utilization of necessary resources for evacuation.

Overall, the utilization of RFID in emergency evacuation helps enhance management, tracking, guidance, and resource optimization.

**5. Discussion**

The proposed approach used the MAS-GiG model combined with four related problems to address the following optimization issues: Route optimization includes investigating ways to optimize the movement paths of individuals/groups during the evacuation process to minimize evacuation time and avoid congestion. For example, determine the shortest route or route with fewer groups of passengers moving to minimize collisions; evacuation plan optimization includes determining the start time of evacuations for each group, the number of evacuees per group, the exit, and efficient resource allocation. For example, which groups of passengers from which areas move towards which safe exit?; and coordination optimization, exploring strategies to optimize the coordination and management of the evacuation process at different levels, from high-level to individual-level management, ensuring collaboration and efficiency during evacuations.

The proposed method is suitable for an evacuation environment with multiple exit routes. The management and monitoring of the crowd are dispersed, leading to high evacuation efficiency as demonstrated by experimental results. In the experiment within the scope of a lounge, the evacuation results for scenario 1 are as follows: the total evacuation time for the area with exit E1 is 95 s, area with exit E2 is 107 s, and area with exit E3 is 133 s. Therefore, the total evacuation time for lounge 5 is 133 s. The evacuation results for scenario 2 are as follows: the total evacuation time for the area with exit E1 is 197 s, area with exit E2 is 0 s, and area with exit E3 is 133 s. Therefore, the total evacuation time for lounge 5 in scenario 2 is 197 s. In this case, since there is a fire near exit E2, passengers cannot move to a safe place through exit E2. The evacuation results for scenario 3 are as follows: the total evacuation time for the area with exit E1 is 189 s, area with exit E2 is 159 s, and area with exit E3 is 0 s. Therefore, the total evacuation time for lounge 5 in scenario 3 is 189 s. In this case, since there is a fire near exit E3, passengers cannot move to a safe place through exit E3.

In this study, we have improved the research method in reference [2] in two aspects: group formation and used one MAS-GiG model for the application. The purpose is to evaluate the research and select a better method. The experimental results between the two methods are equivalent. This indicates that group formation and the number of MAS-GiG models used for the application do not significantly alter the results. The experimental results of the two methods within the scope of four lounges are illustrated as follows: The departure hall consists of four lounges, from Lounge 4 to 7, with each lounge having

its evacuation plan. The total evacuation time for the departure hall is equal to the total evacuation time of the lounge with the longest evacuation time.

In the experiment with four lounges, the evacuation results for scenario 1 are as follows: The total evacuation time for lounge 4 is 159 s, lounge 5 is 197 s, lounge 6 is 190 s, and lounge 7 is 194 s. Therefore, the total evacuation time for the entire departure hall in scenario 1 is 197 s. For the method described in the paper [2], the total evacuation time for Lounge 4 is 157 s, lounge 5 is 199 s, lounge 6 is 192 s, and Lounge 7 is 190 s. Therefore, the total evacuation time for the departure hall using the method described in paper [2] for scenario 1 is 199 s. Similarly, the total evacuation time for the departure hall in scenario 2 is 206 s for the proposed method and 205 s for the method in the paper [2]. The total evacuation time for the departure hall in scenario 3 is 205 s for the proposed method and 207 s for the method in the paper [2].

There are several limitations in the proposed method of combining the MAS-GiG model with four related problems for optimization in emergency evacuation scenarios. Scale limitations: We tested the proposed method in the context of a departure hall of an airport consisting of four lounges, with 300 agents in each lounge. The proposed method may face difficulties when applied to larger and more complex evacuation environments. Evaluation and measurement: For optimization solutions in emergency evacuations, evaluating and measuring effectiveness is an important challenge. Measurement may require careful consideration of criteria and performance indicators, including decision making time, plan deployment time, response time, and safety for evacuees. In the paper, we only consider the total evacuation time and compare it with the method in reference [2]. Scalability and applicability: The proposed method may require adjustments to fit specific situations and the unique requirements of each evacuation system. This can pose challenges in scaling the method and applying it to different scenarios.

However, these limitations also present opportunities for further development and improvement of the method. Future research can focus on addressing these limitations by exploring enhancements in performance, reliability, and feasibility.

## 6. Conclusions

The study proposes an optimization method for emergency evacuation using the MAS-GiG model combined with four related problems. The proposed method has several advantages, such as:

- Using the MAS model allows interaction and cooperation between agents during the evacuation process. This improves manageability and optimizes the evacuation process by dividing tasks and providing information to each agent.
- Solving optimization problems, such as route optimization, evacuation plan optimization, and evacuation coordination optimization.
- The proposed method is suitable for environments with multiple exits. We improve the formation of groups and utilization of a MAS-GiG model in the application. The evacuation results of the proposed method are comparable to the research method in the paper [2].

The proposed method solves the optimization problem in emergency evacuation for acceptable results rather than providing an absolute optimal solution. This method has focused on research and resolving optimization problems to achieve minimum evacuation time, minimize congestion, and avoid collisions among individuals in the crowd.

In the future, we will further develop and improve upon the limitations addressed in this paper, particularly focusing on the human factors involved in the crowd evacuation process. This includes studying the psychology and behavior of individuals during evacuations. Expanding the scope of application as the research focuses on applying the MAS-GiG model and optimization problems in various emergency evacuation scenarios. This includes not only evacuations in case of fire incidents but also other emergencies such as tsunamis and floods. In addition, we will improve the proposed method by implementing RFID technology to leverage its benefits.

**Author Contributions:** Conceptualization, M.O.; Methodology, D.T.H.H., H.T.T.H. and M.O.; Software, D.T.H.H. and H.T.T.H.; Formal analysis, H.T.T.H.; Data curation, D.T.H.H. and H.T.T.H.; Writing—original draft, D.T.H.H.; Writing—review & editing, M.O.; Supervision, M.O. All authors have read and agreed to the published version of the manuscript.

**Funding:** This research received no external funding.

**Data Availability Statement:** Data is unavailable due to privacy limitations.

**Conflicts of Interest:** The authors declare no conflict of interest.

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
