# Peer review of "Combining MAS-GiG Model and Related Problems to Optimization in Emergency Evacuation"

_computers, doi:10.3390/computers12060117_

Round 1
Reviewer 1 Report
Dear Authors,
thank you for your contribution to enhance emergency evacuation! The model and its empirical case study are strong and professional parts of the study, however the readability, scientific soundness and quality of presentation have to be improved:
· Avoid to use references in the abstract and reformulate to include also results and conclusions.
· The aim to observe emergency evacuation needs to be derived from an extended introduction (provide more background information, describe the existing problem setting and your research approach including concrete research questions to solve it) and comprehensive literature review (list and discuss relevant literature: e.g.: https://dl.acm.org/doi/abs/10.5555/3586210.3586239, https://doi.org/10.1016/j.ejor.2014.02.054, https://doi.org/10.1002/fam.2310, https://doi.org/10.1023/A:1015339216366)
· The clarity of the paper suffers under the three structuring levels sometimes only covering one short paragraph (e.g., 3.2.4 …). Limit the headlines to two levels and combine small sections.
· The discussion as well as the conclusion need to be decisively improved to cover relevant literature, discuss the limitation of the study and provide additional managerial insights, further learnings, reflections, interpretations and future research perspectives.
Author Response
Hãy xem phần Ä‘ính kèm

Reviewer 2 Report
In "Combining MAS-GiG Model and Related Problems to Optimization in Emergency Evacuation", the authors present the a method to solve the problem of evacuation of people from buildings to a safe locations. They apply this method to evacuating the 1st floor departure hall of Danang International Aiport.
The authors describe their implemented method and its applicability to emergency evacuation with in good detail. The experimental setup is introduced very well, with assumptions clearly highlighted (walking speeds, evacuee condition, ...). The resulting experiments are introduced in increasing complexity, thereby testing the proposed methods. While the results in evacuation time may not be much better than reference [2], it is arguable that a) minutes matter in evacuation scenarios and b) it is a good test for method correctness that the results are not wildly different than previous data established with other methods.
The paper was an interesting read with good illustrations throughout. The only minor criticism in presentation pertains to (b) in Figures 5 through 7 which may be unnecessary and removing them would maybe allow bigger representations of the - more interesting - simulation interfaces.
Reviewer 3 Report
Update the following comments for the study "Combining MAS-GiG Model and Related Problems to Optimization in Emergency Evacuation".
1. Delete citation from Abstract.
2. Add a section "Literature review" after the Introduction section and before the description of the problem Section.
2. Image quality is not reading-friendly.
3. Provide a notation table to understand equations.
4. The discussions section requires detailed discussions of results, compared to other situations. Compare the study with the following scenarios:
a. Compare the study for the fuzzy environment (A bi-objective robust possibilistic cooperative gradual maximal covering model for relief supply chain with uncertainty).
b. Provide a stability analysis (An optimization technique for national income determination model with stability analysis of differential equation in discrete and continuous process under the uncertain environment) of the solution.
c. Explain the usefulness of RFID for emergency evacuation (Optimized radio-frequency identification system for different warehouse shapes).
5. The conclusions section is very short (first paragraph). Provide some concrete conclusions, derived from the study.
6. Add recent references (from 2023).
Round 2
Reviewer 3 Report
The revision improves the study. Update the following concerns:
1. Update the reference section with the references from 2023 (a minimum of 4~5 references).
2. Comparison and explanations are good. Amend the following comparison and RFID usefulness in the revised paper.
a. The study based on the MAS-GiG model to resolve optimization in emergency evacuation and the study on a bi-objective robust possibilistic cooperative gradual maximal covering model for relief supply chain with uncertainty are two different research studies that address optimization problems in different contexts.
c. Explain the usefulness of RFID for emergency evacuation (Optimized radio-frequency identification system for different warehouse shapes).
